# S2DB-mmWave YOLOv8n: Multi-object detection for millimeter-wave radar using YOLOv8n with optimized multi-scale features

Mengqi Yuan[1,2], Yajing Yuan[1,3]*, Xiangqun Zhang[1,3], Zhenghao Zhu[2], Chenxi Zhao[1], Xiangqian Gao[4], Genyuan Du[1,3]*

**1** School of Information Engineering, Xuchang University, Xuchang, China, **2** School of Information Engineering, North China University of Water Resources and Electric Power, Zhengzhou, China, **3** Henan International Joint Laboratory of Polarization Sensing and Intelligent Signal Processing, Xuchang, China, **4** Henan Shengshi Hengxin Technology Co., Ltd, Xuchang, China

* yyjxcu@126.com (YY); dugy@xcu.edu.cn (GD)

## Abstract

Millimeter-wave (mmWave) radar has become an important research direction in the field of object detection because of its characteristics of all-time, low cost, strong privacy and not affected by harsh weather conditions. Therefore, the research on millimeter wave radar object detection is of great practical significance for applications in the field of intelligent security and transportation. However, in the multi-target detection scene, millimeter wave radar still faces some problems, such as unable to effectively distinguish multiple objects and poor performance of detection algorithm. Focusing on the above problems, a new target detection and classification framework of S2DB-mmWave YOLOv8n, based on deep learning, is proposed to realize more accuracy. There are three main improvements. First, a novel backbone network was designed by incorporating new convolutional layers and the Simplified Spatial Pyramid Pooling - Fast (SimSPPF) module to strengthen feature extraction. Second, a dynamic up-sampling technique was introduced to improve the model's ability to recover fine details. Finally, a bidirectional feature pyramid network (BiFPN) was integrated to optimize feature fusion, leveraging a bidirectional information transfer mechanism and an adaptive feature selection strategy. A publicly available 5-class object mmWave radar heatmap dataset, including 2,500 annotated images, were selected for data modeling and method evaluation. The results show that the mean average precision (mAP), precision and recall of the S2DB-mmWave YOLOv8n model were 93.1% mAP@0.5, 55.8% mAP@0.5:0.95, 89.4% and 90.6%, respectively, which is 3.3, 1.6, 4.5 and 7.7 percentage points higher than the baseline YOLOv8n network without increasing the parameter count.

**Data availability statement:** All relevant data are within the manuscript and its Supporting Information files.

**Funding:** This work was supported by the Henan Province Key R&D Special Project (No. 241111212500) and the Henan Province Key R&D and Promotion Special (Technology Tackling Key) Project (No. 232102210181). The funders had no role in study design, data collection and analysis, decision to publish, or preparation of the manuscript. Thanks.

**Competing interests:** The authors have declared that no competing interests exist.

## 1. Introduction

Frequency-Modulated Continuous Wave (FMCW) radar has gained significant traction in industrial inspection systems and Advanced Driver Assistance Systems (ADAS), particularly for obstacle detection applications in both indoor and outdoor surveillance scenarios. This adoption surge primarily stems from its inherent advantages, including cost-effectiveness and reliable operational capabilities under adverse atmospheric conditions such as haze, smog, blizzards, and particulate-laden environments. In the autonomous driving domain, object recognition technologies employing red, green, blue (RGB) cameras, mmWave radar, and light detection and ranging (LiDAR) have been widely adopted in automotive ADAS and are increasingly integrated into diverse vehicle platforms, including construction machinery, passenger cars, and commercial trucks [1]. Construction machinery vehicles, for instance, typically operate in dust-laden environments with significant airborne particulates (e.g., sand and mud) or under optically challenging conditions such as nighttime operations, inclement weather, or lens contamination. Under these circumstances, ADAS systems must maintain high-precision real-time detection of personnel and objects. However, the visibility and detection performance of RGB cameras deteriorate substantially in extreme environments, necessitating the adoption of alternative sensing modalities such as mmWave radar and LiDAR [2]. Notably, driven by 5G-enabled advancements, mmWave radar has become more cost-effective than LiDAR, significantly enhancing its practical appeal for automotive applications. In surveillance applications for both indoor and outdoor environments, RGB cameras remain the predominant solution; however, their recognition performance degrades significantly under poor illumination, bad weather, and privacy-sensitive scenarios [3]. Moreover, the detection capability is further impaired in the conditions like smoke and haze presence. Collectively, FMCW radar demonstrates greater adaptability to extreme working conditions compared to RGB cameras and LiDAR and higher detection accuracy compared to ultrasonic and infrared sensors [4].

As most of the objects detected using FMCW radar have a micro-Doppler signature, this makes the classification of objects possible. The classical approach to perform these operations, both indoor and outdoor, involves the use of Constant False Alarm Rate (CFAR) thresholds on radar processed signals [5].The method dynamically calculates the threshold by analyzing the ambient noise power surrounding the radar echo signal and determines the presence of a target based on the computed threshold, who significantly enhances target detection probability and makes it more adaptable to complex cluttered environments. However, in non-uniform environments, the detection performance of the CFAR algorithm deteriorates rapidly. In most cases, the CFAR algorithms are difficult to achieve a correct solution to complex identification tasks. Over time, the CFAR algorithm has been modified and improved several new variants, such as Cell Averaging CFAR (CA-CFAR) [6], Order Statistics CFAR (OS-CFAR) [7], and Greatest of CFAR (GO-CFAR) [8]. All current CFAR algorithms perform target detection through a reference window on radar map. However, the use of reference windows reduces detection efficiency and induces

model mismatch issues. Recent studies have applied Machine Learning (ML) algorithms to process the collected radar data, where data-driven approaches learn complex nonlinear relationships between noise and targets to adapt to dynamic environments, while automatically extracting multidimensional features to capture subtle target variations. Experimental results demonstrate that the ML-based approach exhibits high robustness with respect to the traditional CFAR thresholds in noisy scenarios.

With the progressive breakthroughs in machine learning and computational power enhancements from hardware innovations, low-cost millimeter-wave radar systems integrated with deep learning architectures have found applications in gesture recognition [9], human imaging and tracking [10], and indoor mapping [11]. The inherent differences between millimeter-wave radar data and conventional camera imagery necessitate specialized representation and processing methods to optimize deep learning performance in radar applications. Currently, learning from radar data in point cloud format has been extensively studied [12,13]. For instance, [12] proposed a semantic segmentation network for radar point clouds, and [13] adapted PointNets for 2D target detection using radar point clouds. However, point cloud generation often relies on filtering and thresholding techniques to eliminate background clutter and noise, which can lead to information loss due to hard-coded filtering algorithms. To address the information degradation, radar data can be transformed into range-angle-Doppler dimensional heatmaps [14], thereby fully utilizing the signal characteristics inherent to millimeter-wave radar returns.

Initially, the most common ML techniques used for target recognition were based on Convolutional Neural Networks (CNN) [15]. Jiang et al. [16] streamlined the target detection workflow in range-azimuth maps by jointly analyzing range and Doppler dimensions of millimeter-wave radar data. They conducted a comparison experiment that demonstrated the effectiveness of CNN in such tasks. However, the experimental data are simulated and not tested in real scenarios. Today, real-time object detection methods, such as You Only Look Once (YOLO), have been effectively integrated into millimeter-wave radar signal processing for latency-sensitive applications. Gupta et al. [17]developed a dataset comprising range-azimuth heatmaps of targets detected by FMCW radar, subsequently employing a Darknet53-based YOLOv3 architecture for robust target classification across diverse operational scenarios and dynamic target variations. Lamane et al. [18] proposed a hybrid framework integrating FMCW radar, YOLOv7, and Pix2Pix architectures to improve detection accuracy. This method employs Pix2Pix for dataset denoising in range-azimuth heatmaps, followed by training an enhanced YOLOv7 model on the refined thermal representations. Experimental results demonstrate the improvement in detection performance, but also reveal inadvertent suppression of large-scale targets and small objects during the denoising phase, suggesting potential information loss in extreme target size conditions. Kosuge et al. [19] employed a 2D multiple-input multiple-output (MIMO) radar system achieving an imaging effect comparable to RGB sensors. The data from the sensors was fused and fed into the YOLOv3 model for target detection and classification. The experiments demonstrated that the method maintains viable detection capabilities under visual information scarcity, while persisting privacy concerns inherent to the methodology and suboptimal detection performance for low-resolution small targets remain challenging. Kim et al. [20] proposed a radar-to-image conversion method by YOLOv2 network with range-azimuth heatmaps, demonstrating the feasibility of cross-modal adaptation for target detection. Michela et al. [21] achieved robust detection without background suppression through direct fusion of radar data cubes with YOLOv3 frameworks, though the method validation focused primarily on single-target scenarios, leaving multi-object detection capabilities unverified. Zhang et al. [22] proposed a multi-algorithm classification framework utilizing dual-input range-azimuth-Doppler and range-velocity heatmaps processed through YOLOv4, followed by Cartesian coordinate transformation for spatial localization. While existing studies, as noted by Tao et al. [23], predominantly rely on radar-camera fusion for classification tasks, this work investigates millimeter-wave radar as a standalone sensor, aiming to validate its capability for high-precision multi-target detection in complex scenarios.The advantages and disadvantages of the aforementioned related methods are listed in Table 1.

As the images of mmWave radar contain heatmap of objects which has indistinct boundaries or shape, it makes the objects difficult to be separated from the background to recognize. To address the issue, we present a multi-object

**Table 1. Advantages and Disadvantages Analysis of Related Methods.**

| Methods | Advantages | Disadvantages |
|---------|-----------|---------------|
| CFAR | Strong adaptability and Real-time capability | Not suitable for non-uniform backgrounds andLacks contextual understanding |
| [15,16] | Simplifies detection workflow and validates CNN effectiveness | Simulated data only, not validated in real-world scenarios |
| [17] | Achieves wide-angle 180° panoramic observation through radar rotation | Larger errors for small targets or distant targets |
| [18] | Improves detection accuracy after denoising | Denoising suppresses large/small targets and potential information loss |
| [19] | High-quality radar imaging, suitable for fusion with RGB data | Privacy concerns and poor detection of low-resolution small targets |
| [20] | More accurate recognition of large targets like vehicles | Confusion in direction judgment when target speed is slow or point cloud is sparse |
| [21] | Robust detection without background suppression | Validation focused on single-target; multi-object detection unverified |
| [22] | Combination of 3D and 2D detection heads | Not yet supported for real-time embedded deployment; requires optimization and lightweight structure |
| [23] | Adaptable to different environments and scenarios | Requires significant computational resources to handle multimodal data |

detection framework for mmWave radar thermograms based on an improved YOLOv8n architecture, which specifically optimized for radar thermal data through enhanced multi-scale feature extraction mechanisms. The principal contributions of this work can be summarized as follows:

1. Advanced convolutional layers integrated with a Simplified Spatial Pyramid Pooling - Fast (SimSPPF) structure are introduced into the backbone network, replacing conventional convolutional operations and the Spatial Pyramid Pooling - Fast(SPPF) module. This configuration mitigates gradient vanishing while reducing information loss, thereby significantly enhancing both feature extraction capabilities and network robustness—particularly when processing low-resolution radar heatmap data.

2. To mitigate the loss of fine-grained information during upsampling in YOLOv8n, we propose a novel upsampling module. By adaptively fusing multi-level feature representations, this design enables more precise image detail reconstruction and substantially improves object localization accuracy.

3. The neck network incorporates a BiFPN module to optimize multi-scale feature fusion, thereby improving detection performance for objects at varying scales.

4. The proposed S2DB-mmWave YOLOv8n model achieves 93.1% mAP@0.5 and 55.8% mAP@0.5:0.95 on the mmWave radar range-azimuth heatmap dataset, surpassing the baseline by 3.3% and 1.6% respectively, with concurrent precision and recall rates of 89.4% and 90.6%, which demonstrates significant improvements in classification accuracy and overall detection performance for mmWave radar systems.

## 2. Materials and methods

### 2.1. Introduction to YOLOv8n model

YOLOv8n represents an advanced evolution in object detection algorithms, building upon the successes of its predecessors in the YOLO series. The YOLOv8n network architecture consists of three primary components: the Backbone, the Feature Enhancement Network (Neck), and the Detection Head. Feature extraction within the backbone is performed by a

combination of Conv, C2f and SPPF modules. In particular, the C2f module introduced in YOLOv8n enhances the efficiency of feature extraction and lays the foundation for further exploration and optimization. In the Neck module, YOLOv8n extends the PA-FPN architecture by eliminating specific convolutional layers during the up-sampling stage, thereby improving computational efficiency. The Detection Head incorporates a decoupled design that separates classification and localization branches, effectively resolving the inherent conflict between classification and regression tasks while enhancing overall detection performance. The complete network architecture is depicted in Fig 1.

## 2.2. The S2DB-mmWave YOLOv8n methodology

In order to improve the performance of the model in multi-target detection of mmWave radar heat map and enhance its ability of flexible deployment in complex environments, we propose the **S2DB**-mmWave YOLOv8n model, of which the network architecture is shown in Fig 2. The main contributions of the work include three modifications. Firstly, it is the backbone network architecture optimization, called SimBackbone, which replaces the traditional convolutions with simplified convolution (**S**imConv) [24] module and adopts simplified spatial pyramid pooling–fast (**S**imSPPF) to replace the original SPPF module. Secondly, a new up-sampling technology, **D**ySample [25], is used to replace the up-sampling module of the original YOLOv8n, which can gain more detail feature information of the object to fuse. Thirdly, the **B**iFPN [26], including the bidirectional feature propagation mechanism and the feature weighting strategy, is integrated in the neck part to optimize feature fusion, which address the issue of undifferentiated summation in conventional methods by enabling more effective fusion of multi-resolution feature maps with varying importance.

### 2.2.1 Backbone network improvements.

A. SimConv

The structural design of SimConv is depicted in Fig 3. The key distinction between the SimConv and the original convolution of the YOLOv8n model (Conv) lies in the choice of activation functions: SiLU in Conv versus ReLU in SimConv. ReLU is computationally simpler, making it easier to implement and less susceptible to numerical instability compared to SiLU, which results in faster neural network training and inference.

Moreover, ReLU effectively mitigates the gradient vanishing issue commonly observed with activation functions like sigmoid or tanh, offering a more efficient alternative to the traditional sigmoid function. The mathematical formulation of the ReLU function is presented in Equation (1), and its corresponding function graph is illustrated in Fig 4.

$$f(x) = max(0, x) \qquad (1)$$

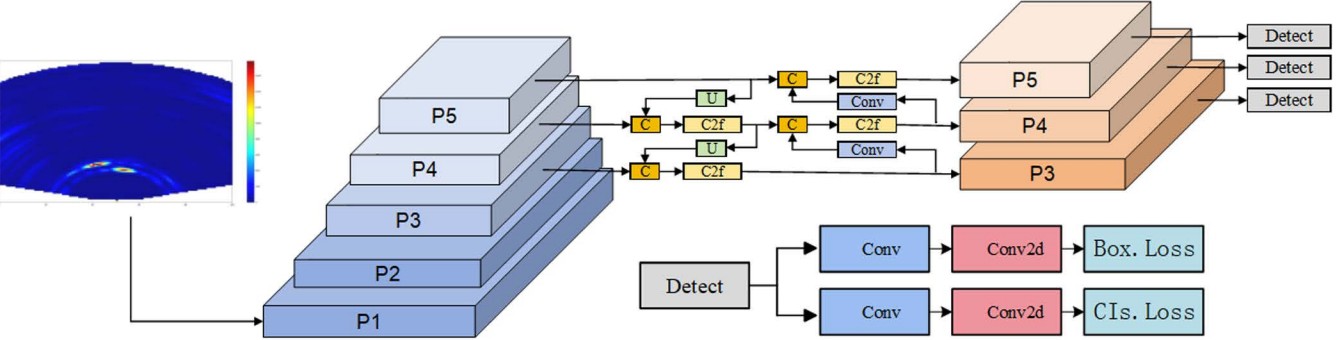

**Fig 1. YOLOv8n network architecture diagram.**

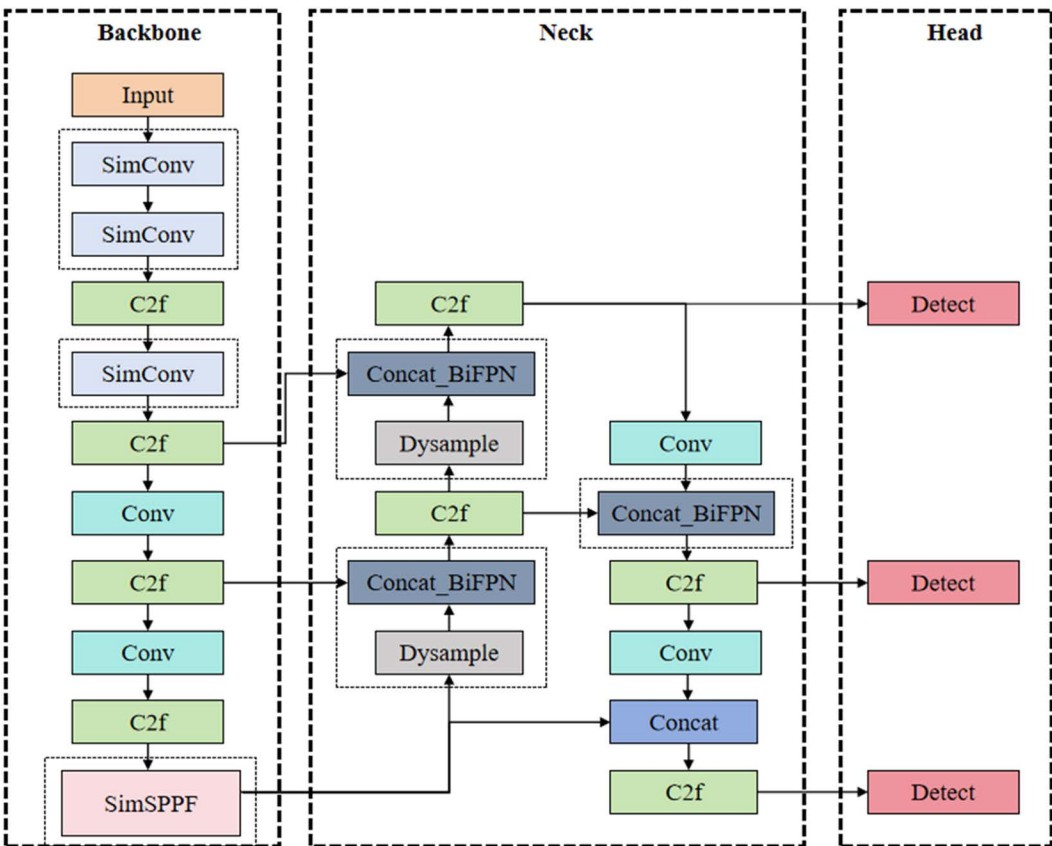

**Fig 2. Improved network structure of the YOLOv8n model.**

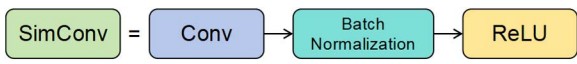

**Fig 3. SimConv structure.**

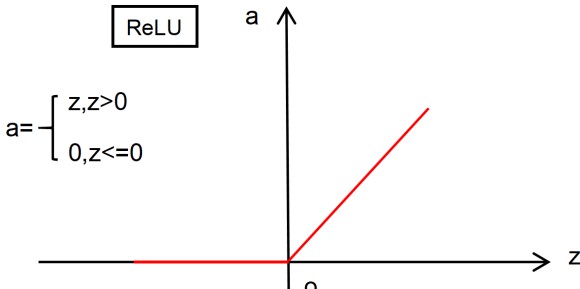

**Fig 4. Image of the ReLU function.**

B.  Pyramid Pooling Layer

Spatial pyramid pooling is an advanced feature fusion technique that integrates multi-scale features by mapping local features into a multi-dimensional space, effectively reducing information loss. In this paper, SimSPPF, an improved spatial pyramid pooling module, is integrated into the YOLOv8n object detection framework to enhance feature extraction. The core improvement of SimSPPF is replacing the Conv module of YOLOv8n model with the SimConv module.

Fig 5 shows the detail structure of SimSPPF, which compresses the input feature map through convolution firstly, and extracts multi-scale features using three sequential MaxPool2D layers with identical configurations, and then the extracted multi-scale features undergo a Concat operation to fuse. The fused features are input in SimConv for upscaling. As the result, the ability of the structure feature integration is enhanced while minimizing information loss during fusion, and the feature expressiveness and detection performance is improved.

**2.2.2.  Up-sampling technique.**  The up-sampling process plays a critical role in feature pyramid network design, where kernel-based dynamic approaches like CARAFE [27], FADE [28], and SAPA [29] have demonstrated improved detection performance. In mmWave radar heatmaps, the presence of numerous small targets and low image resolution poses challenges for up-sampling. Traditional methods like nearest neighbor and bilinear interpolation have limited receptive fields and often lose critical information during multi-scale feature fusion. To address these issues, DySample, a lightweight dynamic up-sampling method, is integrated into the neck part to enhance model robustness to noise while significantly reducing parameter count and computational resource consumption.

The Dysample method workflow is shown in Fig 6, which adapts the group up-sampling strategy that partitions feature maps into multiple independent groups, and each group generates dedicated sampling offsets to minimize inter-feature interference. First, the feature map χ whose size is C×H×W, is resampled by the sampling point generator to generate a set of 2×sH×sW point samples δ, which contains the textual information. Then, the δ and χ are input the grid sample function (grid_sample, as shown in Equation (2)) to conduct a new C×sH×sW feature map χ′ for the further fusion.

$$\chi' = \text{grid\_sample}(\chi, \delta) \tag{2}$$

Fig 7 gives the key sampling point generator process. DySample takes a low-resolution feature map χ as input and first generates a dynamic range adjustment factor through a linear transformation layer (linear1), constraining its value within [0, 0.5] to control the sampling range. Subsequently, another linear transformation layer (linear2) produces an initial offset

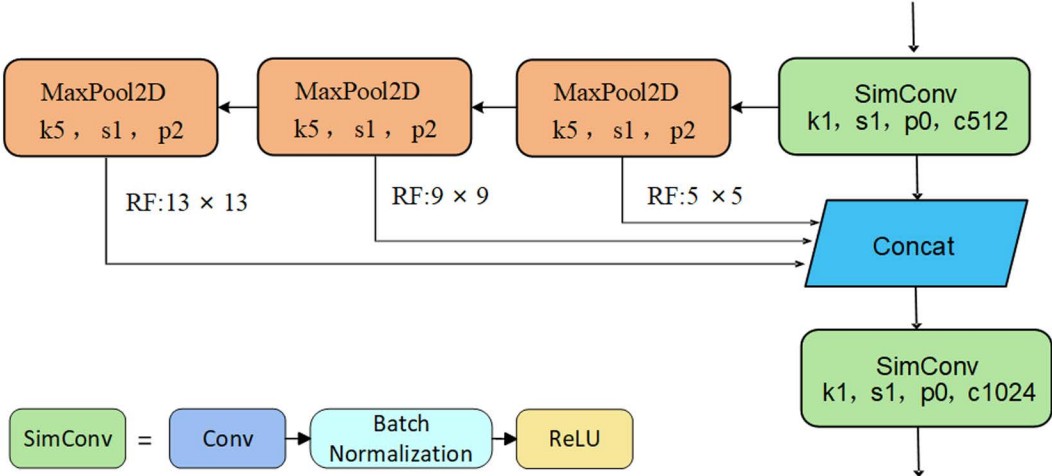

**Fig 5. SimSPPF module structure.**

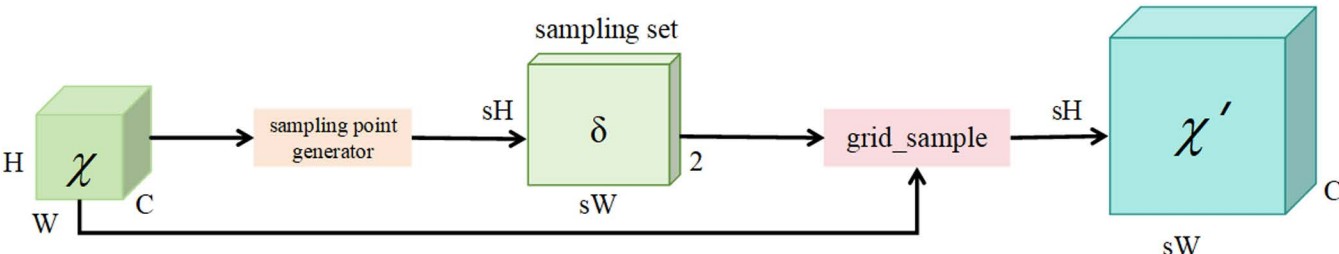

**Fig 6. The sampling workflow diagram of the Dysample module.**

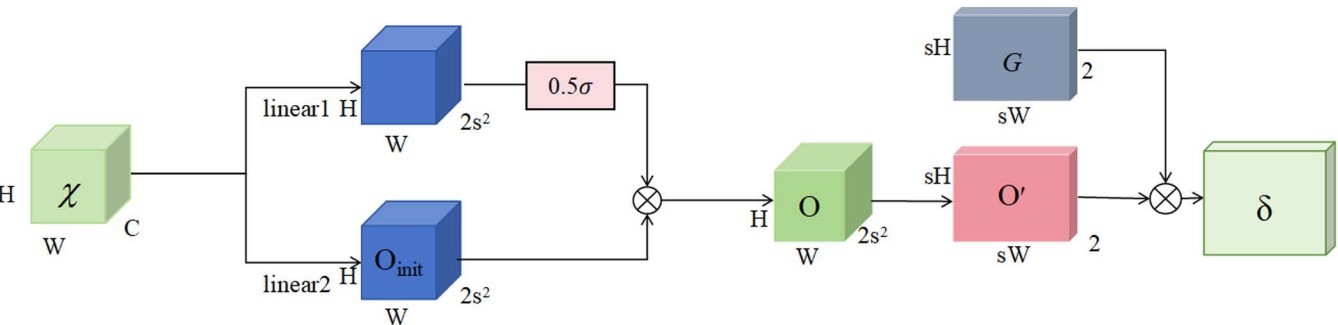

**Fig 7. The structure of the sample point generator in Dysample.**

$O_{init}$. By combining the dynamic adjustment factor with the initial offset $O_{init}$, a dynamic offset is generated. This offset O has dimensions $2s^2 \times H \times W$, and is then rescaled to a $2 \times sH \times sW$ O′, and the final sampling ensemble δ is obtained by summing the offset O′ with the original sampling grid G. Fig 8 displays key parts of the pseudocode within the DySample module.

**2.2.3. Weighted bidirectional feature pyramid networks.** Conventional feature pyramid networks (e.g. PANet [30], whose network is shown in Fig 9) often suffer from information loss and inefficient feature propagation in dealing with multi-scale targets. The use of simple addition (Add) or concatenation (Concat) does not adequately account for the relative importance of each feature during the fusion process.To address these problems, BiFPN, an efficient multi-scale feature fusion module, is introduce in the neck part to improve detection by fusing features from different layers. Fig 10 illustrates the BiFPN network structure, which contains a bidirectional information transfer mechanism,and shows the feature maps of both inputs and outputs. This mechanism fuses feature maps of different resolutions through top-down and bottom-up paths. Feature weights are assigned in a learnable manner to enhance important features and suppress redundant information. By removing individual input edge nodes and creating jump lateral connections between peer nodes, BiFPN effectively enhances feature fusion, reduces redundant computation, and enriches the feature representation of the model.

Using the P6-level feature map (illustrated in Fig 10 as an example), BiFPN generates two fused features: the top-down aggregated feature $P_6^{td}$ and the output feature $P_6^{out}$,formally expressed as:

$$P_6^{td} = \text{Conv} \left( \frac{w_1 \cdot P_6^{in} + w_2 \cdot \text{Resize}\left(P_7^{in}\right)}{w_1 + w_2 + \in} \right)$$

(3)

---

**Algorithm 1** DySample Sampling Function

---

**Require:** Feature map $x \in \mathbb{R}^{B \times C \times H \times W}$, offset map offset $\in \mathbb{R}^{B \times 2gs^2 \times H \times W}$
**Ensure:** Sampled feature map $y \in \mathbb{R}^{B \times C' \times sH \times sW}$

 1: $[B, \_, H, W] \leftarrow \text{shape(offset)}$
 2: $\text{offset} \leftarrow \text{reshape(offset)}$ to $\mathbb{R}^{B \times 2 \times N \times H \times W}$ where $N = g \cdot s^2$
 3: $\text{coords}_h \leftarrow [0.5, 1.5, \ldots, H - 0.5]$
 4: $\text{coords}_w \leftarrow [0.5, 1.5, \ldots, W - 0.5]$
 5: $\text{coords} \leftarrow \text{meshgrid}(\text{coords}_w, \text{coords}_h)$
 6: $\text{coords} \leftarrow$ reshape to shape $[1 \times 1 \times 2 \times H \times W]$
 7: $\text{normalizer} \leftarrow [W, H] \in \mathbb{R}^{1 \times 2 \times 1 \times 1 \times 1}$
 8: $\text{coords} \leftarrow 2 \cdot (\text{coords} + \text{offset})/\text{normalizer} - 1$ ▷ Normalize to $[-1, 1]$
 9: $\text{coords} \leftarrow$ reshape and pixel shuffle to $\mathbb{R}^{B \cdot N \times sH \times sW \times 2}$
10: $\text{input reshaped} \leftarrow$ reshape $x$ to $\mathbb{R}^{B \cdot g \times C_g \times H \times W}$
11: $y \leftarrow \text{grid\_sample}(x, \text{coords})$ with bilinear interpolation, border padding
12: $y \leftarrow$ reshape $y$ to $\mathbb{R}^{B \times C' \times sH \times sW}$
13: **return** $y$

---

**Fig 8. The key parts of the pseudocode within the DySample module.**



**Fig 9. PANet network design.**

$$P_6^{\text{out}} = \text{Conv}\left( \frac{w_1' \cdot P_6^{\text{in}} + w_2' \cdot P_6^{td} + w_3' \cdot \text{Resize}\left(P_5^{\text{out}}\right)}{w_1' + w_2' + w_3' + \in} \right)$$

(4)

where $P_5^{\text{out}}$, $P_6^{\text{in}}$, $P_6^{td}$, $P_6^{\text{out}}$, and $P_7^{\text{in}}$ denote the output feature at level 5, the input feature at level 6, the intermediate feature at level 6, the output feature at level 6, and the input feature at level 7, respectively; Conv represents the convolution operation; Resize refers to upsampling or downsampling operations; w and w' are learnable weight parameters; and $\in$ is set to 0.0001 to ensure numerical stability.

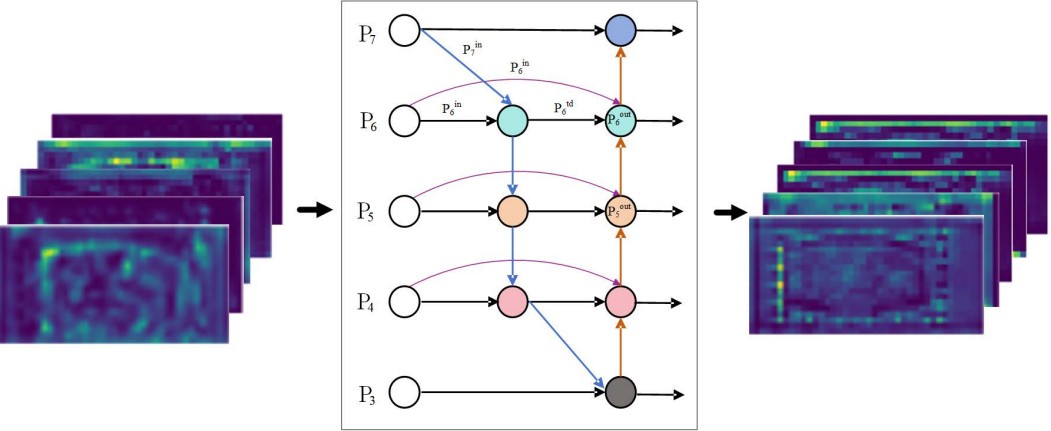

**Fig 10. BiFPN network design.**

## 3. Experiment

### 3.1. The dataset

The dataset used in this paper is based on the public mmWave radar dataset [18], which is acquired using the second-generation single-chip mmWave radar AWR2944, and four radar frames are extracted per second with each frame corresponding to a camera frame. The relevant parameters of the radar are listed in Table 2. The radar data is converted into Cartesian representation and recorded as a heatmap. The heatmap and the camera data are then compared with corresponding labels to ensure compatibility with the YOLO model. The Roboflow tool is used for labeling the target areas, ensuring compatibility with the YOLO model.

The acquisition and labeling of mmWave radar heatmaps is a complex and time-consuming process, which is often limited by insufficient data volume. To address this challenge, data augmentation techniques are applied to the original public dataset to expand the training dataset, which improves the model's generalization and robustness. The data augmentation parameters on the Roboflow platform are set as follows: random brightness adjustment of between −5% to +5% and Salt and pepper noise was applied to 0.14% of pixels. As a result, the dataset is expanded to over 2500 images.

**Table 2. Radar parameters.**

| Items | Parameters |
| --- | --- |
| Framerate | 4Hz |
| Frequency | 77GHz |
| Waveform | FMCW |
| TX antennas | 3 |
| RX antennas | 4 |
| Range resolution | 0.0732m |
| Max range | 9.3743m |
| Azimuth resolution | 14.5° |
| Power consumption | 9W |
| Data rate | <1 Mbps |

## 3.2. Experimental environment and configuration

All experiments were conducted using PyCharm on a 64-bit Windows 11 operating system with Python version 3.8.0. The hardware configuration includes an NVIDIA GeForce GTX 1650 GPU and an Intel(R) Core (TM) i5-9300H CPU @ 2.4GHz. The hyperparameters used for training are listed in Table 3.

## 3.3. Model evaluation metrics

In object detection, precision, recall, and mAP are commonly used to evaluate detectors. Precision measures the proportion of correctly identified positive samples among all those predicted as positive, while recall indicates the ratio of actual positive samples within the predicted set. The formulas for accuracy and recall are presented by Equation (5) and Equation (6):

$$\text{Precision} = \frac{TP}{TP + FP} \tag{5}$$

$$\text{Recall} = \frac{TP}{TP + FN} \tag{6}$$

where TP refers to the detector correctly identifying annotated objects, FP denotes the detector mistakenly predicting background regions as annotated objects, and FN represents the detector incorrectly classifying annotated objects as background regions.

mAP is an important metric for object recognition. It integrates precision and recall into a single value, providing a holistic assessment of a model's accuracy across different thresholds. mAP@0.5 represents the mean average accuracy calculated at an Intersection over Union (IoU) threshold of 50%, and its formula is given in Equation (7). It is often used to assess the overall object localization performance of a model. mAP@0.5:0.95 refers to the mean average accuracy calculated over multiple IoU thresholds ranging from 50% to 95% (with a step size of 5%), which can be described in Equation (8). It provides a more comprehensive assessment of a model's detection performance by considering both localization accuracy and confidence calibration, making it a more rigorous and reliable metric for evaluating object detection models.

$$mAP@0.5 = \frac{1}{C}\sum_{C=1}^{C} AP_C \tag{7}$$

$$mAP@0.5:0.95 = \frac{1}{19}\sum_{t=0.5}^{0.95}\frac{1}{C}\sum_{C=1}^{C} AP_C(t) \tag{8}$$

$$AP = \int_{1}^{0} P(R)dR \tag{9}$$

**Table 3. Table of training hyperparameters.**

| Parameters | Set up |
|---|---|
| Epochs | 100 |
| Learning rate | 0.01 |
| Batchsize | 8 |
| Images size | 640×640 |

In these equations, C is the total number of all categories in the dataset, $AP_C$ is the AP for category C and AP shown in Equation (9) is obtained by calculating the precision values at different recall levels and then averaging them.

This paper uses frames per second (FPS) to evaluate the inference speed of the model. The core idea behind calculating FPS is to measure the total time required for the model to process a certain number of images, and then divide the number of images by this total time to obtain the number of image frames processed per second, as shown in Equation (10).

$$FPS = \frac{n(\text{image})}{t}$$

(10)

The Giga Floating Point Operations per Second (GFLOPs) is a metric for measuring computational performance, indicating the number of billions of floating-point operations that can be executed per second. When describing model performance, GFLOPs is commonly used to assess the computational complexity and efficiency of a model when processing data.

### 3.4. Experimental results and analysis

**3.4.1. Comparison with different models.** To validate the algorithmic advancements of our approach, we conduct comprehensive comparative experiments on the mmWave radar thermogram dataset. The proposed model is rigorously evaluated against state-of-the-art detection architectures including RT-DETR [31], YOLOv9 [32], and YOLOv11 [33]. Table 4 gives the relative performance across all evaluated methods.

High precision alone may cause missed detections, whereas high recall alone may cause false positives. The mAP metric addresses this trade-off by harmonizing both precision and recall, thereby providing a comprehensive performance measure. As shown in Table 4, the proposed S2DB-mmWave YOLOv8n model achieved the best performance in terms of mAP, which improves mAP@0.5 by 3.3% over YOLOv8n and 2% over YOLOv11n, achieving significant performance gains compared to the YOLO series models. Compared with Pix2Pix+YOLOv7-PM across varying scales, though S2DB-mmWave YOLOv8n exhibits a 3% reduction in precision, but this is accompanied by a significant decrease of 89.1M in model parameters and other performance metrics are notably enhanced. The S2DB-mmWave YOLOv8n model achieves

**Table 4. Comparative test results.**

| Model | mAP @0.5 (%) | mAP @0.5:0.95 (%) | Precision (%) | Recall (%) | Parameters (M) | FPS | GFLOPS |
|---|---|---|---|---|---|---|---|
| RT-DETR | 89.9 | 53.5 | 89.2 | 85.4 | 31.3 | 90.0 | 103.5 |
| YOLOv5n | 88.1 | 51.0 | 79.6 | 83.9 | 2.6 | **125.1** | 7.1 |
| YOLOv6n | 85.1 | 47.5 | 74.2 | 81.5 | 4.2 | 85.3 | 9.8 |
| YOLOv7-PM | 90.1 | 49.5 | 89.2 | 84.1 | 71.5 | 81.9 | 189 |
| PiX2PiX +YOLOv7-PM | 91.8 | 52.5 | **92.4** | 83.9 | 92.3 | 75.6 | 231.2 |
| YOLOv9-c | 86.9 | 47.4 | 88.0 | 76.9 | 25.3 | 35.0 | 103.6 |
| YOLOv11n | 91.1 | 56.3 | 84.3 | 83.9 | 2.6 | 82.0 | 6.3 |
| YOLOv12n | 90.5 | 54.9 | 87.5 | 83.0 | **2.5** | 99.1 | **5.8** |
| YOLOv8n -Baseline | 89.8 | 54.2 | 84.9 | 82.9 | 3.2 | 89.2 | 8.1 |
| S2DB-mmWave **YOLOv8n** | **93.1** | **55.8** | 89.4 | **90.6** | 3.2 | 89.3 | 8.1 |

high accuracy with a computational complexity of 8.1 GFLOPs, positioning it in the lower-middle range among comparative baseline modes. This indicates well-preserved feature extraction capabilities without substantial computational overhead. The model attains a competitively high FPS, demonstrating real-time processing efficiency. Furthermore, its efficient architecture design ensurs favorable scalability and hardware compatibility, enabling deployment across diverse operational scenarios. Experimental results demonstrate that S2DB-mmWave YOLOv8n gains the superior object detection capabilities compared with other models across multiple metrics on the millimeter-wave radar heatmap dataset.

**3.4.2. Ablation experiments.** To validate the effectiveness of our work in multi-object detection for mmWave radar heatmaps, we conducted ablation experiments on three key components: SimBackbone, BiFPN and DySample. Five experimental configurations were designed by progressively integrating into the YOLOv8n framework. The contribution of each enhancement to overall detection performance is shown in Table 5.

The ablation results confirm that each proposed enhancement contributes positively to the overall performance. According to the data in Table 5, Experiment 1 uses the original YOLOv8n model, achieving 89.8% mAP@0.5, 54.2%mAP@0.5:0.95, 84.9% precision, and 82.9% recall. Experiment 2 introduced the BiFPN module to optimize the multi-scale feature fusion by leveraging contextual information across different feature levels, enhancing the model's accuracy with mAP@0.5 and precision improving by 1.8% and 3.4% respectively, compared to Experiment 1. In Experiment 3, which replaced the original up-sampling with Dysample, shows increases in mAP@0.5 and mAP@0.5:0.95 by 0.5% and 0.9% respectively, indicating that DySample enhances detail restoration, preserves depth consistency in planar regions, and effectively handles gradual depth variations. Experiment 4, after replacing the backbone network with SimBackbone, the model's mAP@0.5, precision, and recall all improved by 1.5%, 2% and 1.5% respectively, suggesting that the SimBackbone further improves multi-scale feature integration while mitigating gradient vanishing issues. By enhancing computational efficiency and simplifying implementation, it strengthens the model's feature extraction accuracy. In Experiment 5, BiFPN and Dysample were added simultaneously. By replacing the fixed sampling method of BiFPN with DySample, the up-sampling process is dynamically adjusted, and more detailed information is preserved. The results show that these added modules lead to 2.5, 0.3, 1.7 and 6.2 percentage points increases in mAP@0.5, mAP@0.5:0.95, precision and recall respectively, improving detection accuracy. Compared to the baseline model, the proposed model not only achieves optimal detection performance, with increases in mAP@0.5, mAP@0.5:0.95, precision, and recall of 3.3%, 1.6%, 4.5%, and 7.7% respectively, but also maintains the FPS and GFLOPs, ensuring efficiency and performance.

Figs 11–14 illustrates the gradual improvement in performance metrics across the entire enhancement process, emphasizing the contribution of each component to the model's detection capability. To provide a clearer view, the zoomed-in section highlights the performance changes between the 80th and 100th training epochs.

**3.4.3. Visualization results analysis.** Figs 15 and 16 depict the recognition effects of the baseline and improved models on the mmWave heatmap dataset, respectively. As observed from the figures, the improved model enhances performance in recognizing all target scales, with particularly significant improvements in small target detection and

**Table 5. Results of ablation experiments.**

| NO. | BiFPN | Dysample | Sim Backbone | mAP @0.5 (%) | mAP @0.5:0.95 (%) | Precision (%) | Recall (%) | FPS | GFLOPs |
|---|---|---|---|---|---|---|---|---|---|
| 1 | × | × | × | 89.8 | 54.2 | 84.9 | 82.9 | **89.3** | **8.1** |
| 2 | √ | × | × | 91.6 | 53.4 | 88.3 | 85.3 | 48.1 | 8.1 |
| 3 | × | √ | × | 90.3 | 55.1 | 83.8 | 86.3 | 89.3 | 8.1 |
| 4 | × | × | √ | 91.3 | 54 | 86.9 | 84.4 | 87.7 | 8.1 |
| 5 | √ | √ | × | 92.3 | 54.5 | 86.6 | 89.1 | 87.0 | 8.1 |
| 6 | √ | √ | √ | **93.1** | **55.8** | **89.4** | **90.6** | 89.3 | 8.1 |

**Fig 11. mAP@0.5 results.**

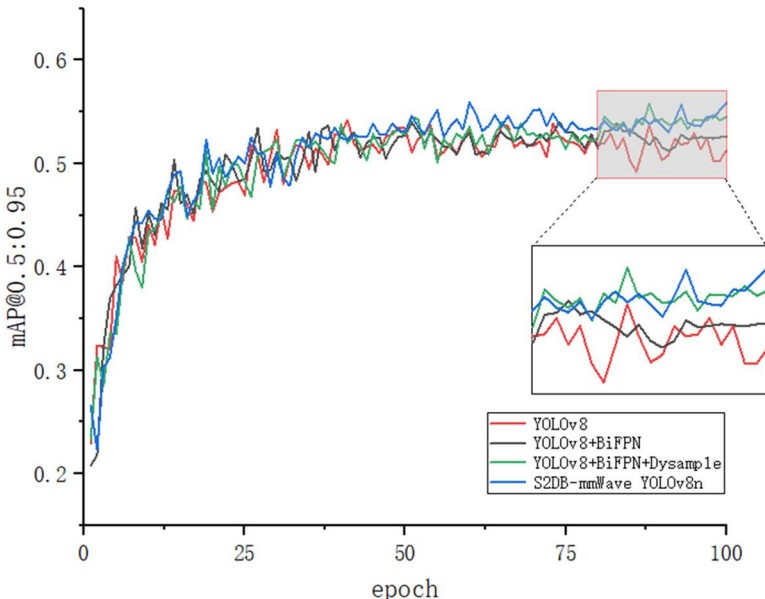

**Fig 12. mAP@0.5:0.95 results.**

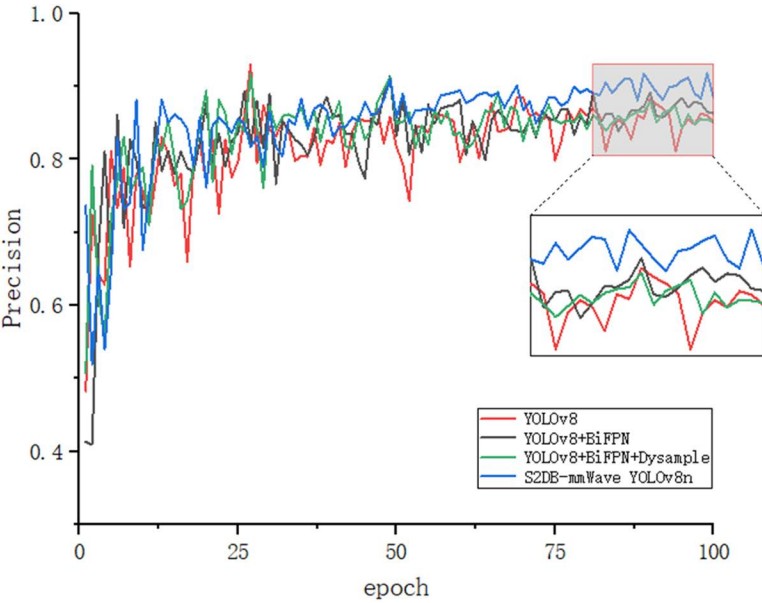

**Fig 13. Precision results.**

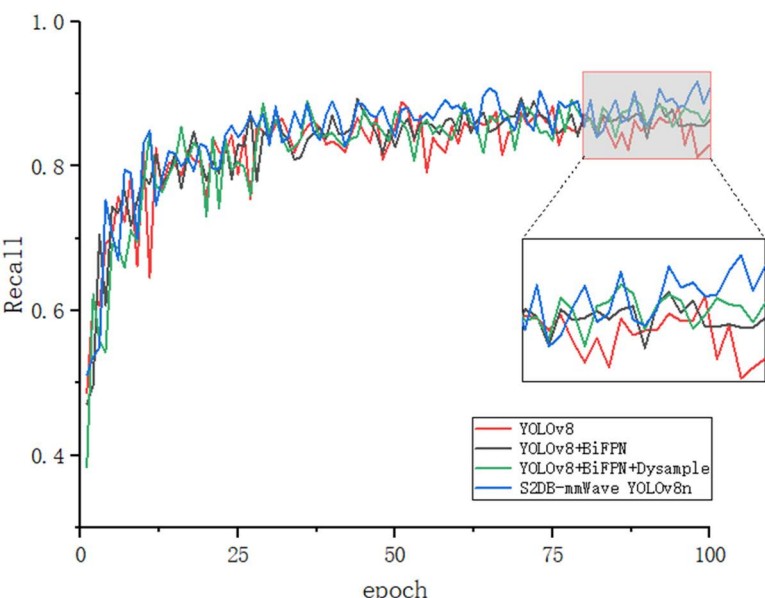

**Fig 14. Recall results.**

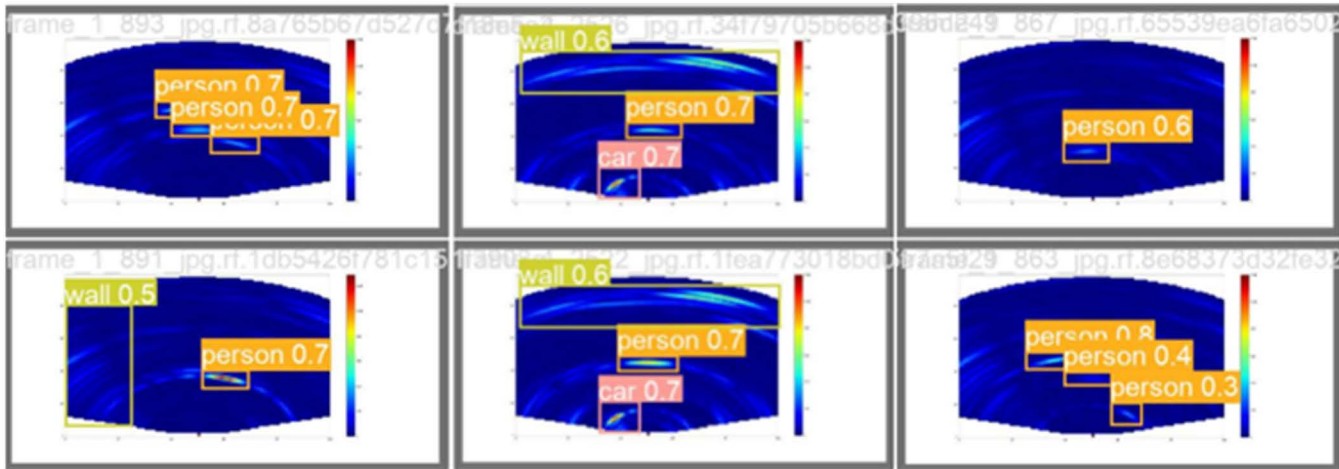

**Fig 15. Plot of baseline model output results.**

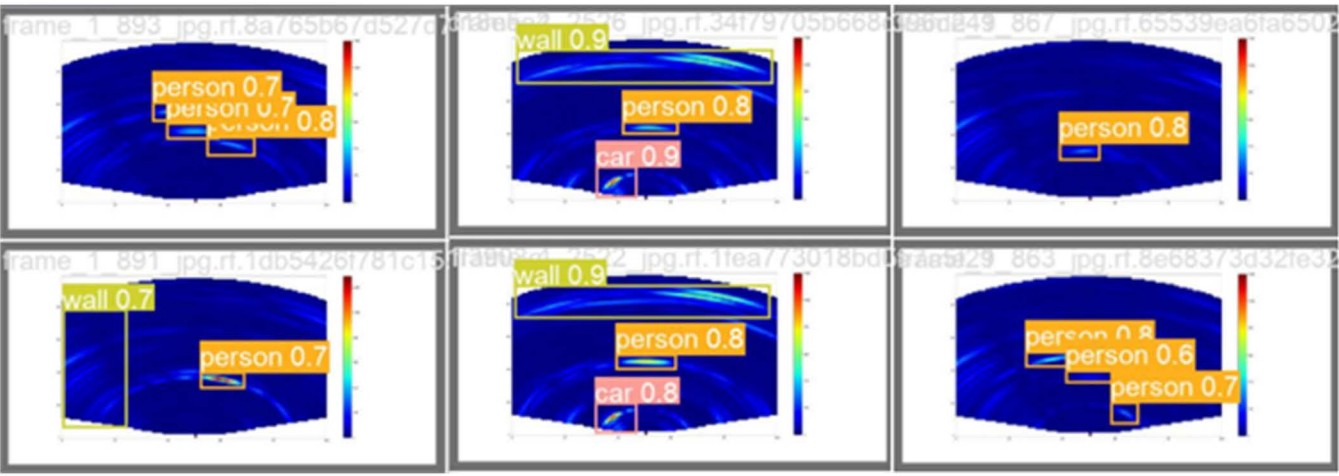

**Fig 16. Plot of improved model output results.**

localization precision. The dual validation through both visual comparisons and quantitative metric improvements (as shown in Fig 11–14) conclusively demonstrates that the proposed network achieves substantially enhanced robustness in addressing inherent multi-scale detection challenges of mmWave radar data.

The confusion matrix provides a comprehensive evaluation of model performance across different target categories, with columns representing ground truth labels and rows indicating model predictions. This visualization effectively reveals both classification accuracy and common misclassification patterns for each target type. Figs 17 and 18 present the confusion matrices for the baseline and improved models respectively, enabling direct comparison of their recognition capabilities. The comparative analysis demonstrates the superior classification performance and the low misclassification rates of our enhanced model.

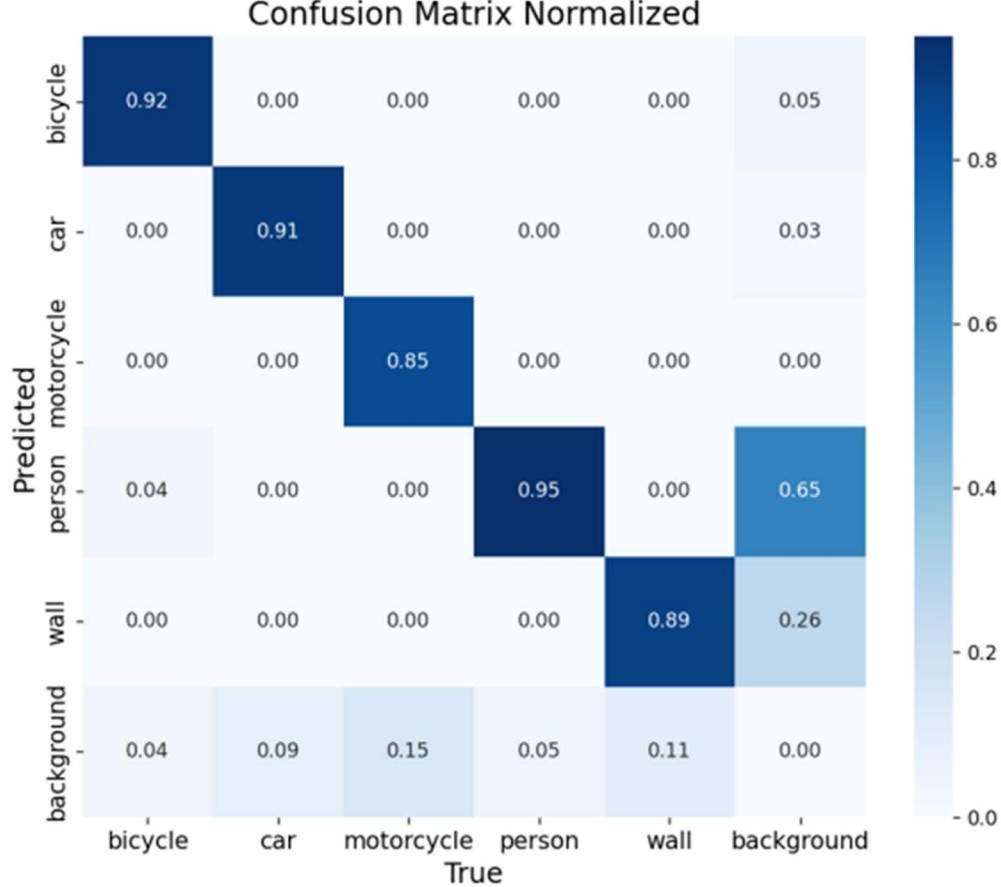

**Fig 17. Diagram of the original model confusion matrix.**

## 4. Conclusions

In this paper, we present an enhanced YOLOv8n model for multi-object detection in mmWave radar heatmaps. The proposed model integrates three key optimizations: (1) a redesigned backbone network to improve feature extraction, (2) DySample for advanced up-sampling, and (3) BiFPN for optimized multi-scale fusion. Extensive evaluations show significant improvements, achieving 93.1% mAP@0.5 and 55.8% mAP@0.5:0.95, with precision of 89.4% and recall of 90.6%, surpassing both the baseline YOLOv8n and comparison models. These advancements provide an effective framework for mmWave radar heatmap analysis, addressing challenges in low-resolution, multi-target detection, especially in challenging environments with dynamic lighting and complex backgrounds.

Due to time, environmental and other constraints, there are still several areas for improvement and further expansion of the research in this paper: (1) exploring attention-based radar-camera fusion to compensate for missing spatial details; (2) adaptive sparse training strategies will further enhance long-range small-target detection.(3) multi-object detection directly on millimeter-wave radar heatmaps still faces significant limitations. Radar heatmaps often suffer from low spatial resolution, severe noise, and object occlusion in dense scenes, making it difficult to distinguish overlapping targets. These challenges lead to degraded detection performance, particularly in scenarios with multiple closely spaced or weak-reflecting

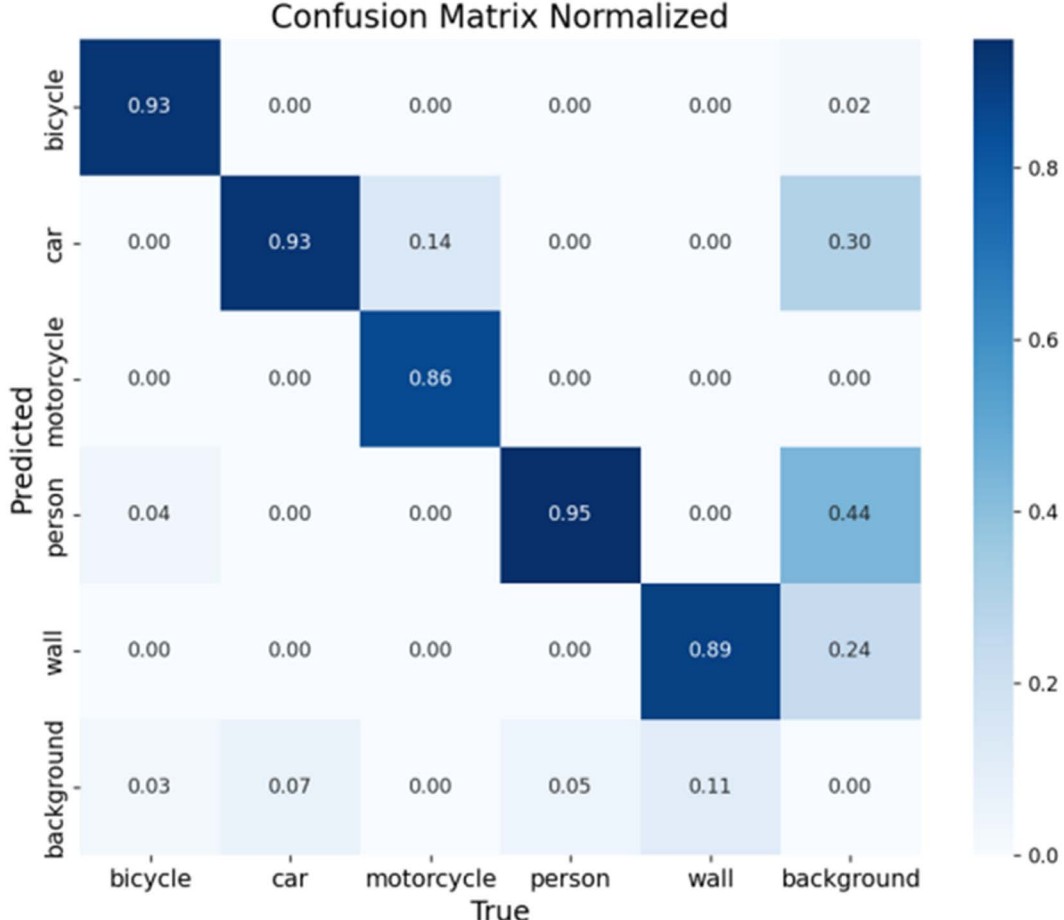

**Fig 18. Confusion matrix of the improved model.**

objects. Therefore, improving heatmap quality, enhancing instance-level feature separation, and incorporating cross-modal priors remain critical directions for future research.

## Author contributions

**Conceptualization:** Yajing Yuan.

**Data curation:** Mengqi Yuan, Zhenghao Zhu.

**Formal analysis:** Mengqi Yuan, Yajing Yuan.

**Funding acquisition:** Genyuan Du.

**Methodology:** Yajing Yuan, Xiangqun Zhang.

**Validation:** Chenxi Zhao, Xiangqian Gao.

**Writing – original draft:** Mengqi Yuan.

**Writing – review & editing:** Yajing Yuan, Zhenghao Zhu.

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
