## [Decision Letter · Decision Letter 0]

20 Jun 2025

PONE-D-25-21299S2DB-mmWave YOLOv8n: Multi-Object Detection for Millimeter-Wave Radar Using YOLOv8n with Optimized Multi-Scale FeaturesPLOS ONE

Dear Dr. Yuan,

Thank you for submitting your manuscript to PLOS ONE. After careful consideration, we feel that it has merit but does not fully meet PLOS ONE’s publication criteria as it currently stands. Therefore, we invite you to submit a revised version of the manuscript that addresses the points raised during the review process.

**ACADEMIC EDITOR:**The paper proposes a target detection and classification framework of S2DB-mmWave YOLOv8n, based on deep learning aiming to gain more accuracy. The authors are suggested to provide a table to list pros and cons of the existing works and then this would lead to the importance of this work. Therefore, list of contributions must be provided. Also, the complexity analysis of the proposed method with comparison to the other state-of-the-art has not been provided in the paper, this must be presented.

We look forward to receiving your revised manuscript.

Kind regards,

Nattapol Aunsri, Ph.D.

Academic Editor

PLOS ONE

Journal Requirements: 

 [This work was supported by Henan Province Key R&D Special Project (No. 241111212500) and Henan Province Key R&D and Promotion Special(Technology Tackling Key) Project(No.232102210181).]. 

[This work was supported by Henan Province Key R&D Special Project (No. 241111212500) and Henan Province Key R&D and Promotion Special(Technology Tackling Key) Project(No.232102210181).]

[This work was supported by Henan Province Key R&D Special Project (No. 241111212500) and Henan Province Key R&D and Promotion Special(Technology Tackling Key) Project(No.232102210181).]. 

5. We note that your Data Availability Statement is currently as follows: [All relevant data are within the manuscript and its Supporting Information files.]

Reviewers' comments:

Reviewer's Responses to Questions

**Comments to the Author**

1. Is the manuscript technically sound, and do the data support the conclusions?

Reviewer #1: Yes

2. Has the statistical analysis been performed appropriately and rigorously? 

Reviewer #1: Yes

3. Have the authors made all data underlying the findings in their manuscript fully available?

Reviewer #1: Yes

4. Is the manuscript presented in an intelligible fashion and written in standard English?

Reviewer #1: Yes

5. Review Comments to the Author

Reviewer #1: The paper presents an interesting and relevant contribution to the field of radar-based object detection by integrating an optimized multi-scale feature enhancement strategy into the lightweight YOLOv8n framework. The following are required to be revised for the improvement of this paper.

- The description of the S2DB module lacks sufficient detail regarding its internal operations and parameters. It would be helpful to include a clearer diagram and a pseudocode-style explanation of how the dual-branch mechanism selects and fuses multi-scale features.

- It would be valuable to evaluate the impact of individual components of the feature optimization strategy (e.g., feature scaling parameters, fusion technique) to better isolate their contributions to overall performance.

- Please clarify whether the mmWave radar dataset used in this work is publicly available or proprietary. If it is not public, consider discussing the possibility of releasing a subset or detailed dataset specifications to enhance reproducibility.

- The paper briefly mentions the computational efficiency of YOLOv8n, but it would be informative to provide precise metrics (e.g., FPS on a given hardware setup, parameter count, FLOPs) for both the baseline and the proposed model to quantify trade-offs between detection accuracy and processing speed.

- Consider adding a dedicated subsection discussing the limitations of the current approach, such as potential challenges in highly cluttered environments or varying weather conditions, and how these might be addressed in future work.

6. PLOS authors have the option to publish the peer review history of their article (what does this mean? ). If published, this will include your full peer review and any attached files.

**Do you want your identity to be public for this peer review?** For information about this choice, including consent withdrawal, please see our Privacy Policy .

Reviewer #1: No

---

## [Author Response · Author response to Decision Letter 1]

26 Jun 2025

Response to Reviewers

Dear Dr. Aunsri and Reviewers,

Thank you for your correspondence and the reviewers’ constructive comments regarding our manuscript titled “S2DB-mmWave YOLOv8n: Multi-Object Detection for Millimeter-Wave Radar Using YOLOv8n with Optimized Multi-Scale Features”(Manuscript ID: PONE-D-25-21299). We sincerely appreciate the time and expertise dedicated to evaluating our work, which has provided valuable insights for strengthening the rigor and clarity of this research.

In this revised manuscript and the accompanying point-by-point response document (uploaded separately), we have carefully systematically all suggestions. Significant revisions include:

(1)In Section 1, we have provided a detailed analysis of the strengths and limitations of the existing technology to offer a comprehensive overview. Additionally, we have clearly listed the main contributions of this study to highlight its significance and innovations.

(2)The details of the model have been elaborated more clearly in Section 2.2. We added feature visualization images(Fig8(b)). These additions enable a more accurate evaluation of the performance enhancement achieved by integrating the module.

(3) Added GFLOPs and FPS to the evaluation metrics section in Section 3.3, and included an analysis of these results in the experiments presented in Section 3.4, and we added comparative analysis with state-of-the-art models (YOLOv12), making the model assessment more comprehensive.

We confirm that every reviewer comment has been thoroughly addressed with corresponding modifications in the manuscript. The track-changes version and clean copy are provided for your convenience.

Academic Editor:

1).The paper proposes a target detection and classification framework of S2DB-mmWave YOLOv8n, based on deep learning aiming to gain more accuracy. The authors are suggested to provide a table to list pros and cons of the existing works and then this would lead to the importance of this work. Therefore, list of contributions must be provided. Also, the complexity analysis of the proposed method with comparison to the other state-of-the-art has not been provided in the paper, this must be presented.

(1)The authors are suggested to provide a table to list pros and cons of the existing works.

Response: Thank you for your valuable comments. I have carefully considered your suggestions and made the following revisions to enhance the clarity and completeness of our manuscript. I Added a detailed analysis of the strengths and weaknesses of existing methods at line 140 in the introduction section. This provides a more comprehensive context for our work.

Revised Section:

Line138-Line139.

Table1.

(2)list of contributions must be provided.

Response: Thank you for your valuable comments. The main contributions of this study are now clearly listed at line 146 in the introduction section. This summary helps readers quickly grasp the key innovations and advancements presented in our research.

Revised Section:

Line146-Line165.

(3)Also, the complexity analysis of the proposed method with comparison to the other state-of-the-art has not been provided in the paper, this must be presented.

Response: Thank you for your valuable comments. I introduced GFLOPs and FPS at line 358 in the evaluation metrics section to provide a more holistic assessment of our model’s performance. Additionally, I have included data comparisons in Table 4 at line 376 and Table 5 at line 402, along with corresponding analyses at line 386 and line 426. To further validate our model’s performance, we have also added a comparative experiment with the latest object detection model, YOLOv12 in Table4. This comparison provides a benchmark against state-of-the-art techniques and demonstrates the competitive edge of our proposed approach. These additions offer a more thorough evaluation and facilitate a better understanding of our model’s efficiency and effectiveness.

Revised Section:

Line358-Line368.

Table4.

Line386-Line393.

Table5.

Line426-Line427.

2). Please ensure that your manuscript meets PLOS ONE's style requirements, including those for file naming. The PLOS ONE style templates can be found at https://journals.plos.org/plosone/s/file?id=wjVg/PLOSOne_formatting_sample_main_body.pdf and https://journals.plos.org/plosone/s/file?id=ba62/PLOSOne_formatting_sample_title_authors_affiliations.pdf.

Response: Thank you for your comment. We have reviewed the manuscript formatting according to the provided template.We have ensured that the manuscript meets PLOS ONE's requirements, including file naming requirements.

Revised Section:

Line13.

3). Please note that PLOS ONE has specific guidelines on code sharing for submissions in which author-generated code underpins the findings in the manuscript. In these cases, we expect all author-generated code to be made available without restrictions upon publication of the work. Please review our guidelines at https://journals.plos.org/plosone/s/materials-and-software-sharing#loc-sharing-code and ensure that your code is shared in a way that follows best practice and facilitates reproducibility and reuse.

Response: Thank you for your comment. We've uploaded the code to github (mengqizh1010/S2DB (github.com)) and made sure it follows best practices and facilitates reproducible and reusable sharing.

4). Thank you for stating the following financial disclosure:[This work was supported by Henan Province Key R&D Special Project (No. 241111212500) and Henan Province Key R&D and Promotion Special(Technology Tackling Key) Project(No.232102210181).].Please state what role the funders took in the study.  If the funders had no role, please state: ""The funders had no role in study design, data collection and analysis, decision to publish, or preparation of the manuscript.""If this statement is not correct you must amend it as needed.Please include this amended Role of Funder statement in your cover letter; we will change the online submission form on your behalf.

Response: The funders had no role in study design, data collection and analysis, decision to publish, or preparation of the manuscript. Thanks.

5). We note that Thank you for stating the following in the Acknowledgments Section of your manuscript:[This work was supported by Henan Province Key R&D Special Project (No. 241111212500) and Henan Province Key R&D and Promotion Special(Technology Tackling Key) Project(No.232102210181).]We note that you have provided funding information that is not currently declared in your Funding Statement. However, funding information should not appear in the Acknowledgments section or other areas of your manuscript. We will only publish funding information present in the Funding Statement section of the online submission form.Please remove any funding-related text from the manuscript and let us know how you would like to update your Funding Statement. Currently, your Funding Statement reads as follows:[This work was supported by Henan Province Key R&D Special Project (No. 241111212500) and Henan Province Key R&D and Promotion Special(Technology Tackling Key) Project(No.232102210181).].Please include your amended statements within your cover letter; we will change the online submission form on your behalf.

Response: Thank you for your comment.We have revised our manuscript according to your instructions. Specifically, we have removed the funding-related text from the Acknowledgments section and updated the Funding section as follows:

Funding :

This work was supported by the Henan Province Key R&D Special Project (No. 241111212500) and the Henan Province Key R&D and Promotion Special (Technology Tackling Key) Project (No. 232102210181).

We have also included these changes in the revised manuscript. Thank you for your understanding and support.

6). We note that your Data Availability Statement is currently as follows: [All relevant data are within the manuscript and its Supporting Information files.]Please confirm at this time whether or not your submission contains all raw data required to replicate the results of your study. Authors must share the “minimal data set” for their submission.

Response: Thank you for your comment. We confirm that all raw data required to replicate the results of our study are available. The original dataset used in this study is publicly available and has been appropriately cited in the manuscript. Additionally, all data augmentation techniques applied to the training set, including the parameters used, are clearly described in the Methods section of the manuscript.Therefore, we believe that our submission includes the minimal data set necessary for replication, as required.Please let us know if any further clarification or additional information is needed.

Thank you once again for your careful review and valuable attention to our work. We truly appreciate your efforts and look forward to the possibility of further collaboration in the future.

Reviewer #1:

Comments: The paper presents an interesting and relevant contribution to the field of radar-based object detection by integrating an optimized multi-scale feature enhancement strategy into the lightweight YOLOv8n framework. The following are required to be revised for the improvement of this paper.

1). The description of the S2DB module lacks sufficient detail regarding its internal operations and parameters. It would be helpful to include a clearer diagram and a pseudocode-style explanation of how the dual-branch mechanism selects and fuses multi-scale features.

Response: Thank you for your meticulous efforts and constructive comments. We have taken your suggestions to heart and have made several improvements to enhance the clarity and comprehensiveness of our work. Specifically, we have provided a more detailed explanation of the module’s details, supplemented by pseudocode to facilitate a better understanding of the module’s functionality. Additionally, we have revised the module’s structure diagram and incorporated relevant formulas to elucidate the internal processes more transparently. We believe these enhancements will significantly improve the readability and interpretability of our manuscript.

Revised Section:

Fig2.

Line255-Line260.

Line262-Line263.

Fig7.

Line288-Line297.

2). It would be valuable to evaluate the impact of individual components of the feature optimization strategy (e.g., feature scaling parameters, fusion technique) to better isolate their contributions to overall performance.

Response: Thank you for your comment. In the ablation study comparison within the experimental section, we have elaborated on the impact of each component of the model on its overall performance. Furthermore, to offer a clearer illustration of the module’s impact, we have incorporated a visual comparison of features before and after feature fusion in Figure 8(b). This enhancement allows for a more intuitive understanding of the module’s role in improving the model’s performance. We trust that these revisions address your concerns and provide a more comprehensive evaluation of our approach.

Revised Section:

Line276-Line278.

Line281-Line282.

Fig8(b).

3). Please clarify whether the mmWave radar dataset used in this work is publicly available or proprietary. If it is not public, consider discussing the possibility of releasing a subset or detailed dataset specifications to enhance reproducibility.

Response: Thank you for your insightful comment. We appreciate your attention to the dataset used in our study. As mentioned in the manuscript, the dataset employed is publicly available and has been properly cited. To bolster the robustness and generalization capability of our model, we applied data augmentation techniques to the training subset of this dataset. The specific parameters utilized in this process are thoroughly documented within the text. We believe these enhancements contribute significantly to the overall reliability and applicability of our model.

4). The paper briefly mentions the computational efficiency of YOLOv8n, but it would be informative to provide precise metrics (e.g., FPS on a given hardware setup, parameter count, FLOPs) for both the baseline and the proposed model to quantify trade-offs between detection accuracy and processing speed.

Response: Thank you for your careful work and kind comments. In response, we have incorporated additional data comparisons of GFLOPs and FPS into the comparative experiments and ablation studies within the experimental section, conducted under specific hardware configurations. This enhancement enables a more comprehensive and detailed evaluation of the different models, providing clearer insights into their respective performance characteristics. We believe these additions significantly strengthen the analysis and address the concerns raised.

Revised Section:

Line358-Line368.

Table4.

Line386-Line393.

Table5.

Line431-Line432.

5). Consider adding a dedicated subsection discussing the limitations of the current approach, such as potential challenges in highly cluttered environments or varying weather conditions, and how these might be addressed in future work.

Response: Thank you for your careful work and kind comments. We fully appreciate and acknowledge your suggestions. In line with the overall structure of the manuscript, we have added an analysis of the limitations of the current method and the proposed solutions for future work in the conclusion section. We believe these additions will provide a more comprehensive perspective and address the concerns raised effectively.

Revised Section:

Line479-Line485.

Finally, we extend our sincere thanks once again for your insightful comments and constructive suggestions on our manuscript. Your feedback has been invaluable in helping us enhance the quality and clarity of our work. We are also deeply grateful for your recognition of the significance and contributions of our research. Your support and guidance are truly appreciated.

We believe these revisions have substantially improved the manuscript's quality and technical soundness. Should further clarifications be required, we remain available to provide additional information.

Thank you once again for your guidance in advancing this work.

Best regards!

Sincerely,

Yajing Yuan, Dr.

Xuchang University

E-mail: yyjxcu@126.com

---

## [Editor Report · Decision Letter 1]

8 Sep 2025

S2DB-mmWave YOLOv8n: Multi-Object Detection for Millimeter-Wave Radar Using YOLOv8n with Optimized Multi-Scale Features

PONE-D-25-21299R1

Dear Dr. Yuan,

We’re pleased to inform you that your manuscript has been judged scientifically suitable for publication and will be formally accepted for publication once it meets all outstanding technical requirements.

Kind regards,

Nattapol Aunsri, Ph.D.

Academic Editor

PLOS ONE
---

## [Editor Report · Acceptance letter]

PONE-D-25-21299R1

PLOS ONE

Dear Dr. Yuan,

I'm pleased to inform you that your manuscript has been deemed suitable for publication in PLOS ONE. Congratulations! Your manuscript is now being handed over to our production team.

Kind regards,

on behalf of

Dr. Nattapol Aunsri

Academic Editor

PLOS ONE